



# A Global Analysis of Dust Diurnal Variability Using CATS Observations

Yan Yu[1], Olga V. Kalashnikova[2], Michael J. Garay[2], Huikyo Lee[2], Myungje Choi[2], Gregory S. Okin[1], John E. Yorks[3], and James R. Campbell[4]

[1]Department of Geography, University of California, Los Angeles, CA, USA
[2]Jet Propulsion Laboratory, California Institute of Technology, Pasadena, CA, USA
[3]NASA Goddard Space Flight Center, Greenbelt, MD, USA
[4]Naval Research Laboratory, Monterey, CA, USA

*Correspondence to*: Yan Yu (yuyan06@gmail.com)

**Abstract.** The current study investigates the diurnal cycle of dust loading across the global tropics, sub-tropics, and mid-latitudes by analyzing aerosol extinction and typing profiles observed by the Cloud-Aerosol Transport System (CATS) lidar aboard the International Space Station. According to the comparison with ground-based and other satellite observations, CATS aerosol and dust loading observations exhibits reasonable quality and insignificant day-night inconsistency, thereby supporting the current analysis of dust diurnal cycle using CATS data. Based on an analysis of variance analytical framework, statistically significant diurnal variability in dust loading is identified over key dust sources, including the Bodélé depression, West African El Djouf, Rub-al Khali desert, and western and southern North America, confirming the previous observation-based findings regarding the diurnal cycle of dust emission and underlying meteorological processes in these regions. Insignificant annual mean dust diurnal variability is identified over the Iraqi, Thar, and Taklamakan deserts. The currently identified significant diurnal cycles in dust loading over the rainforests in Amazon and tropical southern Africa, and drylands in South America and the central Australia, are hypothesized to be driven by enhanced dust emission due to wildfires and enhanced katabatic and frontal winds, respectively.

## 1 Introduction

Dust mobilization and concentration exhibit substantial diurnal variability around the globe (Knippertz and Stuut, 2014), contributing to the radiative (DeMott et al., 2010; Tegen and Lacis, 1996), biogeochemical (Okin et al., 2004), and societal (Al-Hurban and Al-Ostad, 2010; Furman, 2003) impacts of mineral dust on the Earth system. For example, accurate representation of the diurnal variability in dust loading is the key for a realistic simulation of the dust radiative effect on the surface (Osipov et al., 2015; Yue et al., 2009). However, the current state-of-the-science models continue to struggle with the representation of dust diurnal cycle. Based on model simulations using different meteorological drivers and dust source parameterizations, Luo et al. (2004) concluded that ~35-70% variance of dust mobilization is associated with diurnal variability of dust mobilization in the world's major dust sources regions. But the simulated diurnal variations in dust mobilization and concentration were highly sensitive to the choice of meteorological driver dataset and dust source parameterization, suggesting substantial uncertainty in the model-based assessment of dust diurnal variability. Coupled





climate or Earth System models are widely used to study the mobilization, transport, and radiative effects of dust aerosols. However, due to the coarse spatial and temporal resolution of these models, the capability of them at accurately capturing the

diurnal cycle of near-surface winds or convective systems is largely limited, leading to often incorrect simulation of diurnal cycle in dust emission and concentration (Marsham et al., 2011; Todd et al., 2008). Given these uncertainties in the simulated global dust diurnal variability, observational characterization of the variations in dust mobilization and concentration will provide a valuable benchmark for evaluating and constraining such model simulations.

In addition, an improved understanding of dust diurnal cycle is beneficial to biology and geology studies of desert

surfaces as land surface observations are most valuable during low atmospheric aerosol activity. For example, The Earth Surface Mineral Dust Source Investigation, EMIT, is planned to operate from the International Space Station starting in 2021 to determine the mineral composition of the arid land dust source regions of the Earth. EMIT space-borne mission will acquire, validate and deliver updates of surface mineralogy used to initialize Earth System Models through the use of imaging spectroscopy in the Visible to Short Wavelength Infrared (VSWIR) portion of the spectrum sensitive to presence of

atmospheric aerosols. The mission will measure the arid land mineral dust source regions of the Earth, recording the distinct spectral features of the iron oxide, sulfate, clay, and carbonate minerals on the surface, however clear-sky conditions is an important requirement for the mission success. EMIT's atmospheric correction team is working on an evaluation of dust temporal variability to optimize observational opportunities of unobstructed desert surfaces for EMIT target areas. EMIT requirements partially motivated our study.

Observed diurnal cycle in dust mobilization has been documented for several major dust sources, associated with various meteorological processes. For example, over the Bodélé depression, the leading dust source in the globe (Engelstaedter et al., 2006; Koren et al., 2006; Washington and Todd, 2005), dust mobilization are predominantly driven by high surface wind speeds that peak in the morning with the breakdown of the nocturnal low-level jet (Wagner et al., 2016). Over the Western African El Djouf, the second largest dust source in North Africa (Yu et al., 2018), a great portion of dust

storms are caused by strong downbursts associated with deep convection in the afternoon (Fiedler et al., 2013; Heinold et al., 2013). Over the Iraqi desert in the Middle East, summertime dust activation is primarily driven by the strong, persistent Shamal wind, which peaks around local noon with intensified low-level temperature gradient (Yu et al., 2016). Beyond these studies that focused on a specific dust source region, there has been limited global analysis of the observed diurnal variability in dust mobilization and concentration.

Satellite- and ground-based aerosol loading measurements and aerosol type classifications are useful for quantitative assessment of the observed global dust diurnal variability, but have to address several methodological challenges. Sun-synchronous passive satellite instruments, such as the Multi-angle Imaging SpectroRadiometer (MISR) (Diner et al., 1998; Kalashnikova et al., 2005) on the Terra satellite and Moderater resolution Imaging Spectroradiometer (MODIS) on both Terra and Aqua satellites, provide observations of dust aerosol optical depth (DAOD), but only cover several snapshots

during the daytime. Observations from lidar instruments, such as the Cloud-Aerosol Lidar with Orthogonal Polarization (CALIOP) on the polar-orbiting Cloud-Aerosol Lidar and Infrared Pathfinder Satellite Observation (CALIPSO) satellite



(Winker et al., 2009), provide both daytime and nighttime measurements of vertically resolved aerosol extinction and aerosol type information. However, CALIOP only samples two fixed temporal points at each pixel on the Earth, and is therefore insufficient for studying the full diurnal cycle. Geostationary sensors, such as the Advanced Himawari Imager on the

Himawari-8 and -9 satellites (Bessho et al., 2016), Geostationary Ocean Color Imager the Communication, Ocean and Meteorological Satellite (Choi et al., 2018), Advanced baseline Imager on the GOES-16/17 satellites (Schmit et al., 2017), and the Spinning Enhanced Visible and Infrared Imager (SEVIRI) instrument aboard the Meteosat Second Generation satellite (Schepanski et al., 2007), only provide observations over a certain region during the daytime. AErosol RObotic NETwork (AERONET) (Holben et al., 1998) based sun and lunar photometers provide hourly or sub-hourly measurements

of aerosol properties, but have limited spatial coverage only over the sites.

The aforementioned challenges in the observational assessment of global dust diurnal variability are partly addressed by the Cloud-Aerosol Transport System (CATS) lidar aboard the International Space Station (ISS) (McGill et al., 2015). CATS is an elastic backscatter lidar that operated on the ISS for 33 months during February 2015 to October 2017. The ISS 51˚ inclination orbit enables CATS measurements to occur at different local time every overpass, with full diurnal coverage for a

given location within a 60-day period (Yorks et al., 2016). By comparing CATS-derived AOD and aerosol vertical distributions with aerosol properties derived from other ground-base and satellite-based observations such as AERONET, MODIS, and CALIOP, Lee et al. (2018) found reasonable agreements between aerosol observations from CATS and other sensors, thereby verifying that CATS provides an encouraging opportunity for studying aerosol diurnal variability. By examining CATS aerosol observations, Lee et al. (2018) further identified strong diurnal cycle in total AOD over North

Africa, India, and the Middle East, likely attributed to the diurnal variations in dust generation. Several limitations in this preliminary study were noted by Lee et al. (2018). First, although Lee et al. (2018) reported a substantially better agreement between CALIOP and CATS AODs during nighttime, likely due to enhanced solar contamination during daytime that affects the data quality of both instruments (Campbell et al., 2012), this day-night data inconsistency was not accounted for in the assessment of aerosol diurnal variability. Furthermore, Lee et al. (2018) did not formally perform a significance test of the

aerosol diurnal cycle, leading to a potential overinterpretation over regions that have high mean aerosol loading. Therefore, this preliminary global assessment of aerosol diurnal variability using CATS observations motivates a more sophisticated investigation that accounts for potential day-night data inconsistency and explicitly quantifies the significance of dust diurnal variability.

The current study investigates the observed diurnal variability in dust loading over the global tropics, sub-tropics, and

mid-latitudes by examining aerosol extinction and aerosol type observations from CATS. The day-night data quality consistency is assessed through a comparison of daytime and nighttime CATS observations with AERONET sun and lunar data. The spatial variations in CATS-derived DAOD is assured with MISR non-spherical, dust AOD. A statistical approach is undertaken to systematically determine the statistical significance of the dust diurnal variability. The dust diurnal cycle over key regions are discussed along with the driving meteorological processes. The methods, results, and

conclusions/discussion are provided in Sections 2, 3, and 4, respectively.



## 2 Data and Methods

### 2.1 CATS

CATS Level 2 (L2) Version 3-00 5 km Aerosol Profile products (L20_D-M7.2-V3-00_5kmPro, L20_N-M7.2-V3-00_5kmPro) are used in the current study for the entire period of CATS operation on the ISS during February 2015 - October
2017. CATS L2 profile data is provided with 5 km horizontal resolution along track, in 533 vertical levels at 60 m vertical resolution, and at a wavelength of 1064 nm (Pauly et al., 2019). Data at 532 nm is also provided by CATS but not recommended for use due to a laser-stabilization issue (Yorks et al., 2016). Thus, only 1064 nm products are used in the current study. The accuracy of the extinction coefficient has improved from Version 2, which was used in Lee et al. (2018), to Version 3 CATS products, as a result of several improvements in the retrieval algorithms, especially during the daytime.
For aerosol typing, CATS uses layer-integrated 1064 nm depolarization ratio, layer base altitudes and thickness, surface type, and GEOS modeled aerosol species to discriminate dust, smoke, polluted continental, marine, and upper stratosphere/lower troposphere aerosols.

CATS data are quality-assured (QA), mainly following (Lee et al., 2018) with minor modifications. QA thresholds include: 1) Extinction_QC_Flag_1064_Fore_FOV = 0, 2) Feature_Type_Fore_FOV = 3 (aerosol), 3) -10 <=
Feature_Type_Score_FOV <= -2, and 4) Extinction_Coefficient_Uncertainty_1064_Fore_FOV <= 10 $km^{-1}$.

In the current study, both total AOD, as reported in the CATS standard product, and DAOD, as computed from vertical profiles of extinction coefficient and feature/aerosol type, are analyzed. AOD from CATS is compared with AERONET to assess the data quality during both daytime and nighttime. DAOD is defined here as the vertical integral of aerosol extinction coefficient over "dust" (Aerosol Type = 3) or "dust mixture" (Aerosol Type = 4) pixels, thereby reflecting the total dust
loading at each location. The spatial variations in DAOD is evaluated against MISR.  In light of the larger uncertainty associated with a reported AOD of zero (Toth et al., 2018), any AOD or DAOD that equal to zero is ignored in the current analysis, following Campbell et al. (2012). This approach results in ~1000 of DAOD retrievals in any three-hour local time window at each 2˚x2˚ grid cell over the dust source regions, such as North Africa, and fewer than 100 retrievals over remote oceans (Supplemental Figure 1).

### 2.2 AERONET

In order to assess the CATS data quality at both day and night, Version 3 AERONET AOD from both sun and lunar photometers are analyzed. The level 2 (cloud screened and quality assured) daytime (Giles et al., 2019) and level 1.5 (cloud screened) nighttime AOD observations (Barreto et al., 2019, 2016) at the 1020 nm spectrum band are compared with collocated CATS AOD at 1064 nm. Here a "collocated observation" is identified when the CATS orbit passed anywhere in
the ± 0.5 latitude/longitude box of a specific AERONET site within ± 0.5 hour of the corresponding AERONET site observation. Previous observational studies have identified a threshold of 40 km and 3 hours beyond which the spatial and temporal autocorrelation drops below 80% (Anderson et al., 2003; Omar et al., 2013). In the current study, the relatively



broad criteria on spatial and temporal collocation is expected to maintain the spatial and temporal autocorrelation while provide reasonable number of collocated observations, especially for the nighttime comparison (Figure 1). Note that one
AERONET measurement is often associated with multiple CATS retrievals in both space and time. In this case, CATS data is averaged spatially and temporally, resulting in only one pair of collocated and averaged CATS observations for a given collocated incident at each AERONET site.

## 2.3 MISR

In order to assess the validity of CATS DAOD, Version 23, Level 3 MISR nonspherical, dust AOD at 550 nm during
the CATS operation period is analyzed here. The MISR nonspherical AOD fraction is often referred to as "fraction of total AOD due to dust," as dust is the primary nonspherical aerosol particle in the atmosphere, especially over desert regions (Kalashnikova et al., 2005). In light of the narrow swath of MISR and the resulting limited number of collocated observations from CATS and MISR, here global maps of seasonal average DAOD from CATS and MISR are compared, thereby verifying the spatial variations in CATS DAOD. Given the Terra overpass time around 10:30 am local time, only
morning data from CATS (8 am to local noon) are used for the seasonal average. In order to achieve sufficient sampling from CATS, DAOD from both CATS and MISR are aggregated into a 2˚latitude x 2˚ longitude grid.

## 2.4 ANOVA-based significance test of dust diurnal variability

In the current study, the dust diurnal variability at each location and its statistical significance is estimated and tested under an analysis of variance (ANOVA) framework (Fisher, 1992). At each pixel, the $k^{th}$ DAOD observation in local time
window i and season j ($D_{ijk}$), is approximated as a sum of the global, annual mean DAOD (D), an annual mean diurnal term ($d_i$), a seasonal term ($s_j$), a diurnal term varying by season ($ds_{ij}$), and an error term that reflects other factors ($\varepsilon_{ijk}$), namely:

$$D_{ijk} = D + d_i + s_j + ds_{ij} + \varepsilon_{ijk} \quad (1)$$

In order to achieve sufficient sample size, CATS DAOD is aggregated into each 3-hour local time window (0-3, 3-6, etc), in each season [December-February (DJF), March-May (MAM), June-August (JJA), and September-November
(SON)], in each 2˚ latitude  x 2˚ longitude pixel, so that within each combination of pixel, time window, and season, there are at least 50 DAOD observations during 2015-2017.

The statistical significance of the diurnal variability, namely variability in $d_i's$, is determined through an F-test, following the classical two-way ANOVA variance partitioning approach. An F-statistic is constructed as

$$F = \text{Variance of } d_i\text{'s} / \text{Variance of } \varepsilon_{ijk}\text{'s.} \quad (2)$$

Under the null hypothesis that there is no diurnal variability in DAOD at a specific pixel, F follows an F-distribution with degrees of freedom ($df_1$, $df_2$) determined by the number of diurnal time windows (8 in the current study), number of seasons (4), and total number of observations in each pixel (assuming equal to n), with

$$df_1 = 8-1 \quad (3)$$



$df_2 = n - 8 - 4 - 8x4$     (4)

Based on the value of the F-statistic and its degrees of freedom, a p-value can be determined, thereby determining the statistical significance of the diurnal variability. If the p-value is smaller than the pre-determined threshold (0.05 in the current study), it is very likely that the alternative hypothesis that there is diurnal variability is true. In the results and discussion sections, we mainly focus on regions with significant diurnal variability (p-value < 0.05). Note that in order to apply the F-test, the assessed variable is required to follow Gaussian distribution. Therefore, a log-transformation is
performed on the observed DAOD, and the application of the ANOVA framework is based on the logarithm of DAOD.

## 3 Results

### 3.1 Comparison of CATS and AERONET AOD

As an initial assessment of the potential quality inconsistency between daytime and nighttime CATS data, total AOD from CATS is evaluated against AERONET, and the agreement between CATS and AERONET AOD, in terms of temporal
correlation, root-mean-square-error (RMSE), and mean bias, is compared among daytime and nighttime collocated observations at each AERONET site. According to the comparison (Figure 1), there is no significant difference between daytime and nighttime CATS AOD quality. Although nighttime CATS AOD observations exhibit apparently higher mean temporal correlation (0.69), lower RMSE (0.11), and lower absolute bias (+0.04) with respect to collocated AERONET AODs, than those of daytime observations (0.62, 0.13, and +0.07, respectively), these differences are not statistically
significant according to the Student's t-test, likely due to the wide spread of these three metrices among different AERONET sites in both daytime and nighttime comparisons. The currently identified insignificant difference between daytime and nighttime CATS data quality is hypothesized to be partly attributed to 1) small sample size of collocated AERONET-CATS AOD observations, and 2) the difference in quality check levels of the currently applied daytime (cloud-screened and quality-assured) and nighttime (cloud-screened only) AERONET data. In response to the latter hypothesis, future studies are
encouraged to repeat this comparison but with quality-assured AERONET nighttime data when it is released. According to the comparison with AERONET daytime and nighttime data, we hypothesize that the potential day-night data quality inconsistency does not affect the assessment of the full diurnal cycle of dust loading.

Although there is no significant difference between the daytime and nighttime CATS data quality, both daytime and nighttime CATS AOD retrievals appear to underestimate the ground truth from AERONET at high AOD (Figure 2). When
the mean AOD exceeds 0.1, stations with higher mean AOD typically exhibits higher RMSE, lower temporal correlation, and larger magnitude of negative bias, indicating degraded CATS data quality in the presence of high aerosol loading. When the mean AOD exceeds 0.3, the negative mean bias between CATS and AERONET is about -0.15, or 50% of mean AOD with both daytime and nighttime retrievals.



### 3.2 Comparison of CATS and MISR DAOD

According to the comparison with MISR, CATS generally well captures the spatial variations in DAOD, with highest agreement during the boreal winter (Figure 3). Consistent features captured by both MISR and CATS DAOD fields include (1) annually high dust loading over the tropical eastern Atlantic ocean and the seasonally-varying meridional distribution of the maximum dust loading drive by dust transport from North Africa (Yu et al. submitted), (2) seasonally enhanced dust activity over the Middle East, Arabian Peninsula, and Arabian Sea in MAM and JJA associated with active frontal passage (Yu et al., 2015b) and Shamal events (Yu et al., 2016), respectively, and (3) elevated level of dustiness over the northern Pacific ocean in boreal spring due to enhanced dust emission and transport from the Taklamakan and Gobi deserts (Yu et al., 2019). The most pronounced inconsistency between MISR and CATS DAOD fields is the apparent underestimation of dust loading over land, especially over regions with high dust loading, such as North Africa. This apparent underestimation of high dust loadings by CATS is consistent with the underestimation of high aerosol loadings, as discussed in section 3.1.

### 3.3 Global diurnal variability in DAOD

As demonstrated in the global maps of CATS DAOD in each 3-hour local time window (Figure 4), the diurnal variability in dust loading is typically more pronounced over land than over ocean, likely due to the fact that dust over ocean is primarily transported from remote dust sources over land. Over the global terrestrial area, 28% of the 2°x2° latitude/longitude pixels exhibit statistically significant diurnal variability in DAOD ($p<0.05$), according to the ANOVA-based F-test, compared with 2% of the oceanic pixels. In sections 3.3.1-3.3.4, the diurnal cycle of DAOD over North Africa and Middle East, Asia, North America, and the Southern Hemisphere, as well as the underlying meteorological processes are discussed in detail.

### 3.3.1 North Africa and Middle East

Over North Africa and the Middle East, 42% of the terrestrial area exhibit significant diurnal variability in dust loading, mostly corresponding to previously identified dust sources including the Bodélé depression, West African El Djouf, Iraqi desert, and Rub-al Khali desert (Ginoux et al., 2001, 2010, 2012; Prospero et al., 2002; Yu et al., 2013, 2018) (Figure 5). Over the Bodélé depression, the world's leading dust source region, the diurnally maximum dust loading occurs shortly after sunrise, consistent with previous analyses based on geostationary satellite instrument SEVIRI (Chaboureau et al., 2007; Schepanski et al., 2009) and visibility observations (N'Tchayi Mbourou et al., 1997). The Rub-al Khali desert in the central Arabian Peninsula displays a similar diurnal cycle of DAOD with the Bodélé depression, with a morning peak hypothetically also associated with the break of nocturnal low-level jet. Over El Djouf, the second largest dust source in North Africa (Yu et al., 2018), the diurnal cycle in DAOD exhibits a morning peak over the western sub-region, likely associated with the break of nocturnal low-level jet (Fiedler et al., 2013; Schepanski et al., 2009), and an afternoon peak over the eastern sub-region, likely associated with the enhanced deep convection due to surface heating (Heinold et al., 2013). The diurnal





variability in DAOD is less significant over the Iraqi desert than over other key dust source regions in North Africa and the Middle East. Although the mean diurnal cycle in DAOD peaks around local noon, consistent with previous station-based analysis of wind speed and dust storm frequency (Yu et al., 2016), the diurnal variability in DAOD is only significant during the boreal summer with active Shamal (not shown), and thereby weakening the annual mean signal here. Over all the aforementioned key dust sources, the daily minimum DAOD typically occurs at night.

### 3.3.2 Asia


In contrast to the North African and Middle Eastern dust sources, the majority of the terrestrial area in Asia exhibit insignificant dust diurnal variability (Figure 6). Two of the major dust sources in Asia, namely the Thar desert in India and Taklamakan desert in China, both show spatially inhomogeneous and statistically insignificant dust diurnal variability. According to simulations from a regional climate model, the dominant source of dust over the Thar desert varies by season
(Banerjee et al., 2019). While local dust sources provide the majority of dust during the boreal summer monsoon season, the remote sources in the Middle East, Arabian Peninsula, and West Asia contribute comparable amount of dust than the local sources to the northern India during the rest of the year. Since the diurnal cycle of transported dust is less deterministic than locally emitted dust, the overall annual mean diurnal cycle over the Thar desert is less robust than that over the dust source regions in North Africa. Over the Taklamakan desert in China, although the mean range between the diurnally maximum and
minimum DAOD reaches about 20% of the long-term average, quantitatively consistent with a previous AERONET-based assessment at a nearby site (Wang et al., 2004), the diurnal cycle in DAOD is mostly insignificant because the regional dust emission is mainly driven by the frontal passage, which does not have a clear diurnal cycle (Luo et al., 2004).

### 3.3.3 North America

Over North America, about 23% of the terrestrial area exhibits statistically significant dust diurnal variability, although
both the mean and diurnal range of DAOD are much smaller than those over North Africa, the Middle East, and Asia (Figure 7). An afternoon to early-evening peak in DAOD spreads across the southern and western sub-region of the continent, consistent with previous visibility-based and meteorology-based observational analysis, which identified deep convection as the dominant process driving the dust emission across North America (Stout, 2015).

### 3.3.4 Southern Hemisphere


The Southern Hemisphere exhibits generally weak dust loading, especially in terms of diurnally minimum DAOD (Figures 8-10), with significant dust diurnal variability over the Amazon Rainforest, Patagonia in Chile and Argentina, tropical southwestern Africa, and central Australia. Since there have been limited exploration of the diurnal dust cycle over these regions, hypotheses are provided here regarding the underlying driving processes of these diurnal dust variations.

Over the Amazon Rainforest, it has been believed that remote dust sources in North Africa provide key nutrients to
fertilize the Amazon Rainforest through trans-Atlantic dust transport (Kaufman et al., 2005; Koren et al., 2006; Yu et al.,





2015a). However, since the trans-Atlantic dust transport typically takes several days to weeks, it is unlikely that the amount of transported dust displays a clear diurnal cycle. On the other hand, field observations and model simulations have identified a potential contribution from local, post-fire dust emission over vegetated area (Mahowald et al., 2005; Wagenbrenner, 2017; Wagenbrenner et al., 2013). In particular, the observed burned fraction has substantially increased

across the Amazonia and tropical southern Africa during the recent years (Andela et al., 2017), which might be responsible for the small-magnitude, yet statistically significant diurnal variability over the Amazonian and tropical southwestern African rainforests.

Over the majority of Patagonia in South America, the highest dust loading typically occurs after sunset or before sunrise. Dust activation over the mountainous Patagonia is believed to be primarily driven by katabatic winds which

are often caused by surface cooling at night (Bullard et al., 2016). Although a series of modeling studies and geochemical analyses have identified Patagonia as the main dust supplier to the Southern Ocean and Antarctica (Gaiero, 2007; Gili et al., 2016; Tanaka and Chiba, 2006), this region is barely detected by passive satellite sensors, such as MODIS (Ginoux et al., 2012). The absence of Patagonia in the MODIS DAOD-based dust source identification is potentially attributed to the currently identified, statistically significant nighttime peak in DAOD.

Over the central Australia, DAOD generally peaks in the early evening. Dust emission in the central Australia is believed to be driven by frontal passage, similar with the Taklamakan desert in China (Knippertz and Stuut, 2014). However, over central Australia, low-level winds associated with cold fronts intensify with the deformation and convergence in the early evening, driven by the subsidence of the mixing layer (Thomsen et al., 2008), thereby leading to an enhanced dust emission at that time of the day.

## 4 Conclusions and Discussion


Based on the profiles of aerosol typing and extinction observed by Cloud-Aerosol Transport System (CATS) on the International Space Station (ISS), the diurnal cycle of dust loading over the global tropics, subtropics, and mid-latitudes is quantified, with the statistical significance of dust diurnal variability determined through the Analysis of Variance (ANOVA)-based F-test. Based on the comparison between Aerosol Optical Depth (AOD) derived from CATS and Aerosol

Robotic Network (AERONET), daytime and nighttime CATS retrievals exhibit insignificant difference in the data quality, thereby supporting the analysis of full diurnal cycle using CATS data. The spatial variations in dust AOD (DAOD) are reasonably captured by CATS, according to the comparison with the nonspherical, dust AOD from the Multi-angle Imaging SpectroRadiometer (MISR). The analytical framework yields statistically robust findings about the dust diurnal cycle that confirms previous geostationary-based, and ground-based observational conclusions in key dust source regions, including (1)

the Bodélé depression, Rub-al Khali desert, and western El Djouf, which exhibit a morning peak in DAOD driven by the break of nocturnal low-level jet, and (2) the eastern El Djouf, and the southern and western North America, which exhibit an afternoon peak in DAOD, driven by an enhanced deep convection. Insignificant diurnal cycle in DAOD are found over these





dust source regions: (1) the Iraqi desert, where the noon peak in DAOD associated with the enhanced pressure gradient driven by spatially differential heating is only robust in boreal summer, (2) the Thar desert, where the dominant source of dust varies by season, and (3) the Taklamakan desert, where dust emission is primarily driven by frontal passage which does not exhibit a clear seasonal cycle. Over the Southern Hemisphere, three hypotheses are raised regarding the currently identified dust diurnal cycle: (1) post-fire dust emission is likely responsible for the maximum dust loading over the rainforests in Amazon and tropical southwestern Africa, and (2) the early evening peak in DAOD over the central Australia is attributed to the enhanced dust emission by intensified low-level winds with the subsidence-induced convergence.

By comparing the CATS AOD and DAOD with that from AERONET and MISR, respectively, underestimation by CATS in the presence of high aerosol or dust loading are identified here. Previous studies based on Cloud-Aerosol Lidar with Orthogonal Polarization (CALIOP) observations noted that the laser backscatter signal becomes totally attenuated at particulate column optical depths of about 3, so that there are occasions where lidars, such as CATS and CALIOP, cannot measure the full extent of the vertical column in the thick dust layer (Vaughan et al., 2009). Furthermore, the CATS feature detection algorithm creates a gap between the surface and near-surface aerosol base altitude, causing false regions of "clear-air" between the surface and near-surface aerosol layers despite the possible presence of aerosols in this altitude region. CATS does not use an aerosol base extension algorithm, like CALIOP, that detects scenarios when aerosols are present in the bins just above the surface and extends the near-surface aerosol layer base down to the surface (Tackett et al., 2018; Yorks et al., 2019). The complete attenuation and feature detection problems likely lead to underestimation of the near surface dust extinction by CATS over dust source regions, such as the El Djouf, Bodélé depression, and Middle East, underestimation of DAOD as compared with MISR (Figure 3), as well as underestimation of total AOD as compared with AERONET (Figure 2).

The diurnal cycle of dust emission and dust loading over the Southern Hemisphere requires further investigation. The dust sources in the Southern Hemisphere provide key nutrients to the oceans in the Southern Hemisphere, and thereby playing an important role in the global biogeochemical cycle (Mahowald et al., 2011). However, the Southern Hemispheric dust sources received much less research efforts than the Northern Hemispheric dust sources in the past. Beyond the hypotheses regarding the dust diurnal variability and the underlying driving processes raised in Section 3.3.4, influence from potential uncertainty in the CATS aerosol loading and typing retrievals, such as misclassification between dust, smoke, and biological particles (Graham et al., 2003) and presence of aerosols under optically thick cloud layers, needs to be investigated from other sources of observation to validate the currently assessed diurnal cycle in dust loading over the Southern Hemisphere.



**Author Contribution**

YY led the study with inputs from all coauthors. OVK, MJG, JEY, and JRC provided guidance on the satellite data processing. HL and MC contributed to the statistical analysis. GSO helped with the results interpretation. YY prepared the manuscript with contributions from all coauthors.

**Acknowledgements**

CATS L2 aerosol profile data and MISR L3 aerosol data were obtained from the NASA Langley Research Center Atmospheric Science Data Center. AERONET V3 AOD data was obtained from the AERONET website (https://aeronet.gsfc.nasa.gov/). This work was partially performed at the Jet Propulsion Laboratory, California Institute of Technology, under a contract with the National Aeronautics and Space Administration. The authors thank the MISR team for providing facilities and useful discussions.

**Data availability**

The research data used are available at the sources specified in the acknowledgement section.

**Competing interests**

The authors declare that they have no conflict of interest.

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



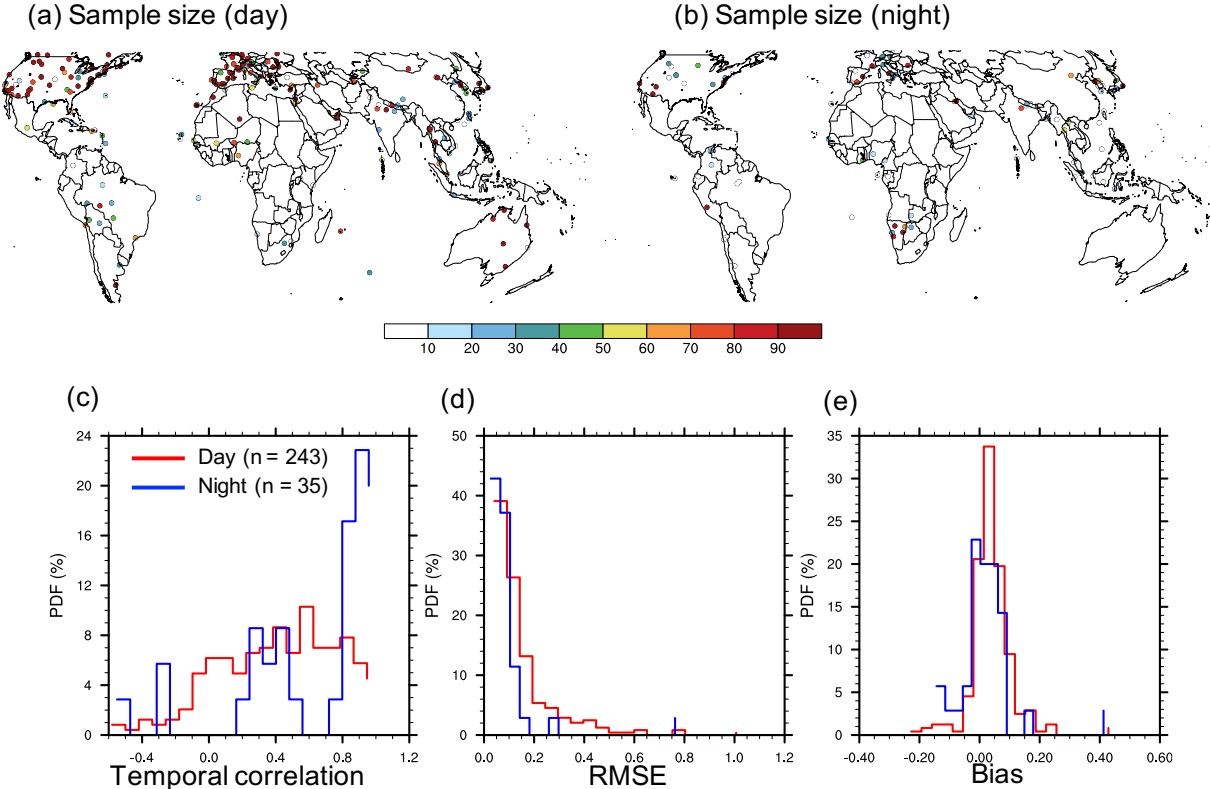

**Figure 1: Comparison of total AOD from CATS (1064 nm) and AERONET (1020 nm) during the local day and night. (a, b) number of collocated observations between CATS and AERONET during (a) daytime and (b) nighttime. (c-e) Probability distribution function (%) of the (c) temporal correlation, (d) root-mean-square-error (RMSE), and (e) mean bias (CATS-AERONET) between collocated AOD observations from CATS and AERONET among all AERONET sites at local day (red curves, n= 243 sites) and night (blue curves, n = 35 sites).**





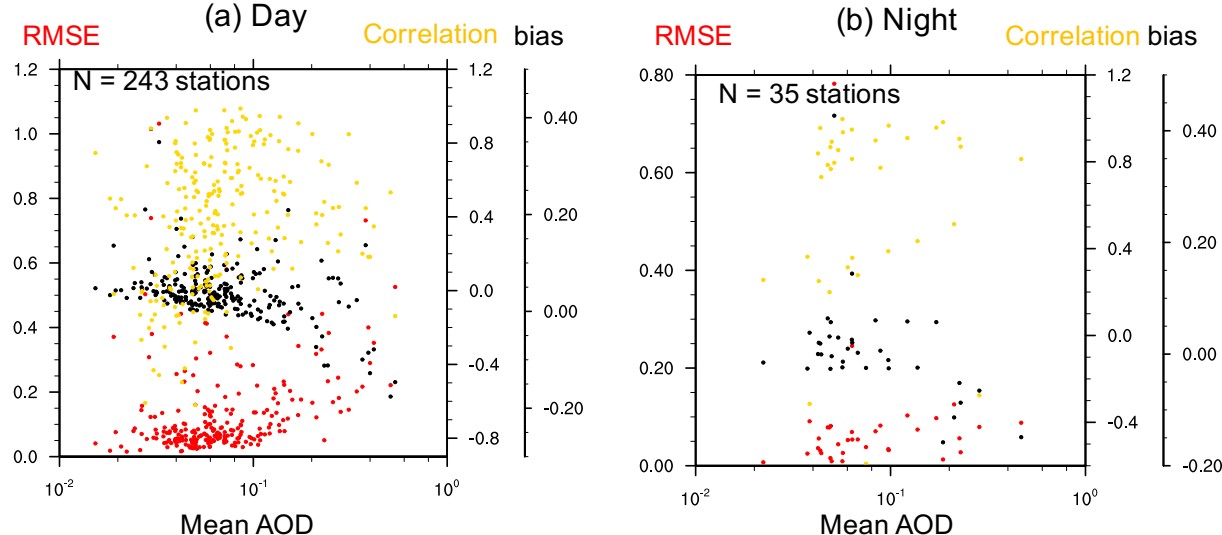

**Figure 2 CATS-AERONET comparison metrics as a function of mean AOD across AERONET sites at local (a) day and (b) night. RMSE, temporal correlation, and mean bias at each AERONET site is presented by a dot in red, yellow, and black, referring to the left, first right, and second right Y-axis, respectively. Mean AOD is obtained from the corresponding AERONET site by averaging all available observations during 2015-2017.**



Figure 3 Comparison of seasonal average, standardized DAOD from (a-d) CATS (1064 nm) and (e-h) MISR (558 nm) in (a, e) December-February (DJF), (b, f) March-May (MAM), (c, g) June-August (JJA), and (d, h) September-November (SON). For a direct comparison, DAOD from each instrument in each season is standardized, namely divided by the 95th percentile of all DAOD observations between 51°N-51°S from that instrument in that season and multiplied by 100. The spatial rank correlation between the seasonal DAOD maps from CATS and MISR are indicated in the corresponding CATS panel.





**Figure 4 Annual-average DAOD from CATS in each 3-hour local time window: (a) 00-03, (b) 03-06, (c) 06-09, (d) 09-12, (e) 12-15, (f) 15-18, (g) 18-21, (g) 21-24.**



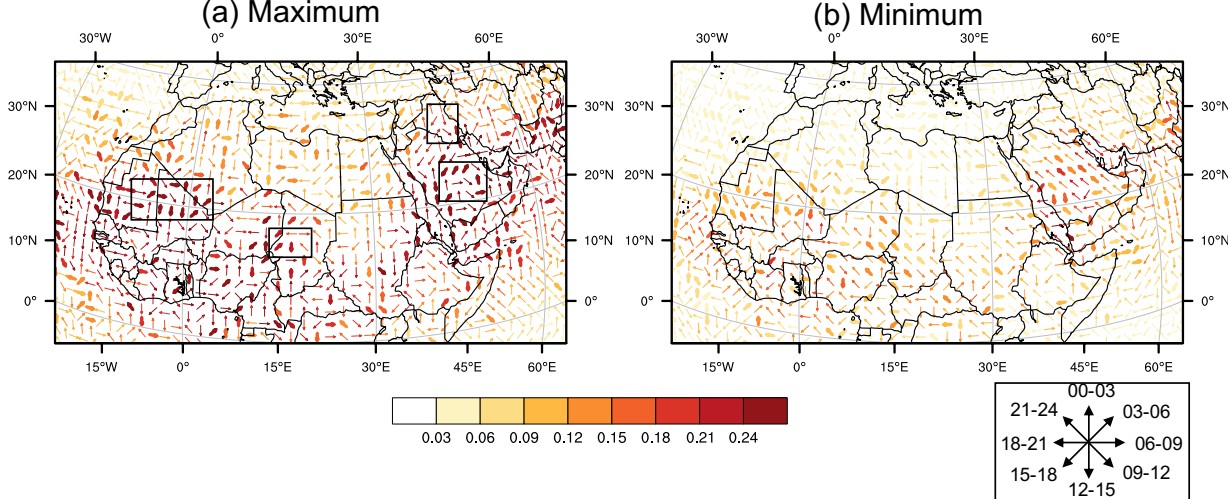

**Figure 5** Diurnally (a) maximum and (b) minimum DAOD across North Africa and Middle East. Direction of the vectors indicates the local time when mean DAOD reaches the (a) maximum and (b) minimum value, with the legend located in the bottom-right corner of the figure. Color of the vectors indicates the (a) maximum and (b) minimum DAOD value. Thick vectors indicate statistically significant diurnal variability ($p < 0.05$) according to the ANOVA-based F-test. The boxes show the location of the Western African El Djouf, Bodélé depression, Iraqi desert, and Rub-al Khali desert from west to east.



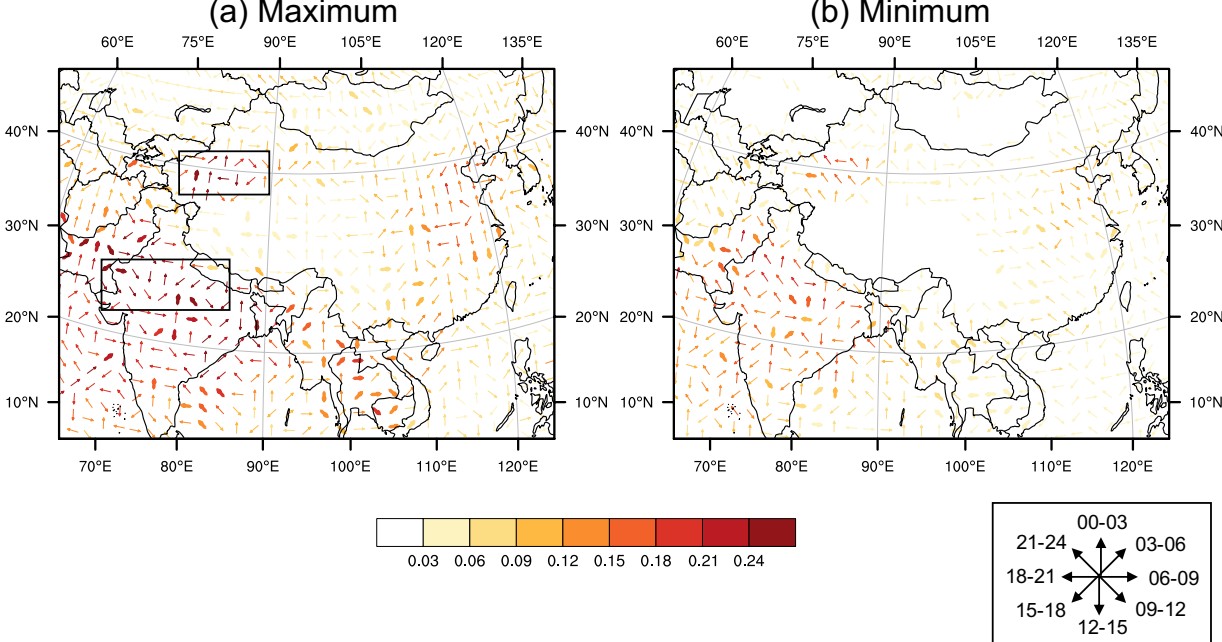

565  **Figure 6 Diurnally maximum and minimum DAOD across Asia. Figure elements are the same as in Figure 5. Boxes indicate the location of the Thar and Taklamakan deserts from west to east.**





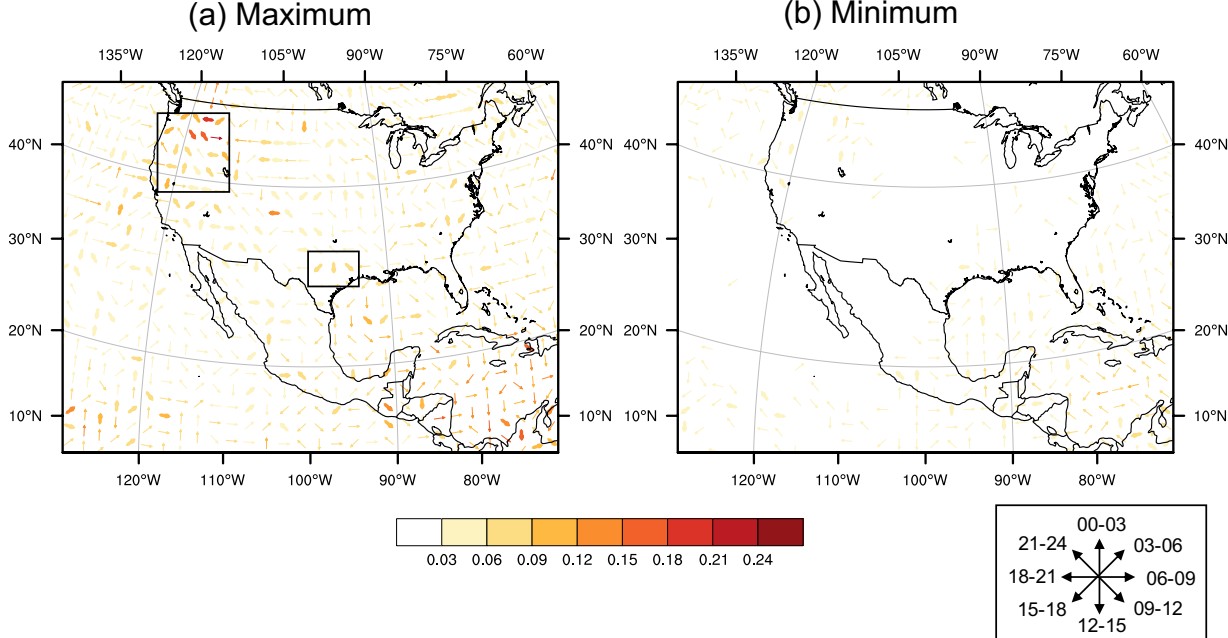

570

**Figure 7 Diurnally maximum and minimum DAOD across North America. Figure elements are the same as in Figure 5. Boxes indicate the dust source regions in the western and southern United States.**



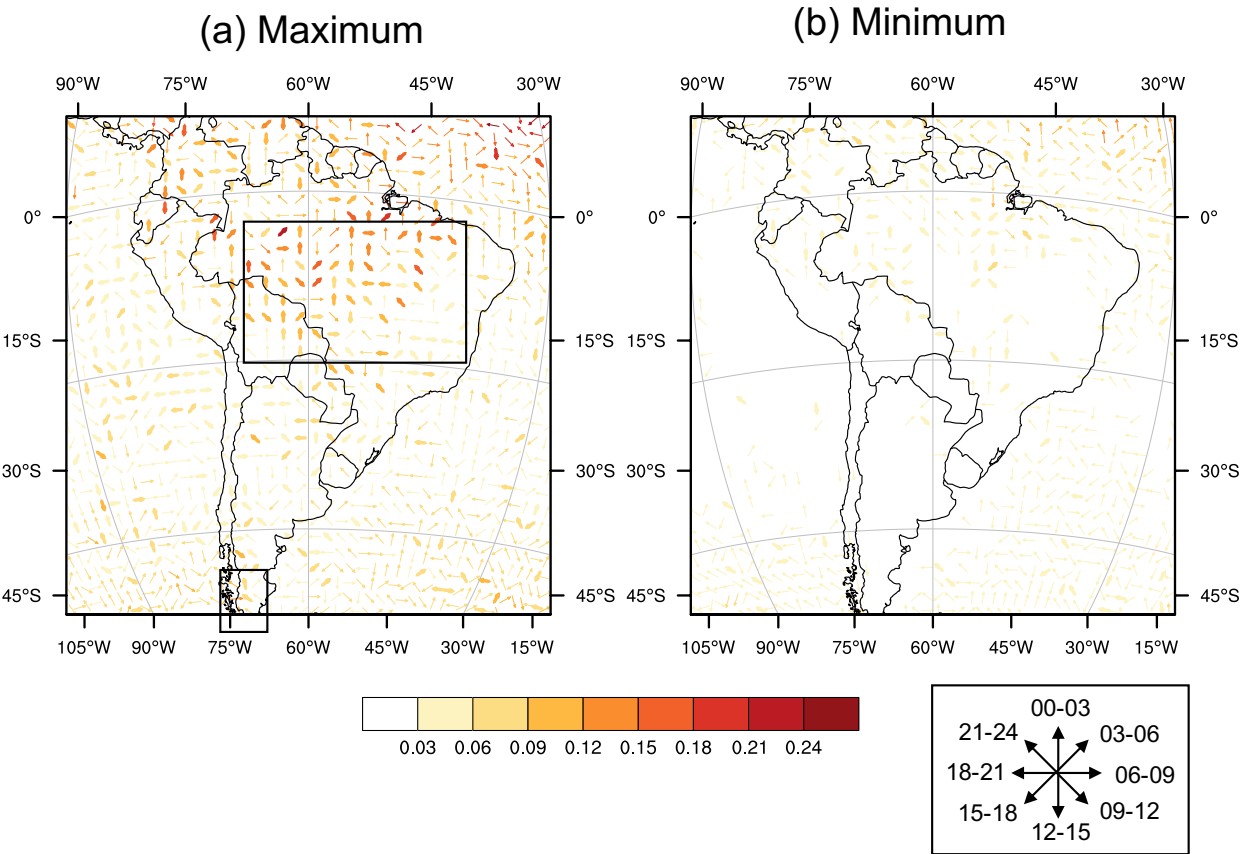

**Figure 8 Diurnally maximum and minimum DAOD across South America. Figure elements are the same as in Figure 5. Boxes**
575 **indicate the Amazon and Patagonia regions from north to south.**



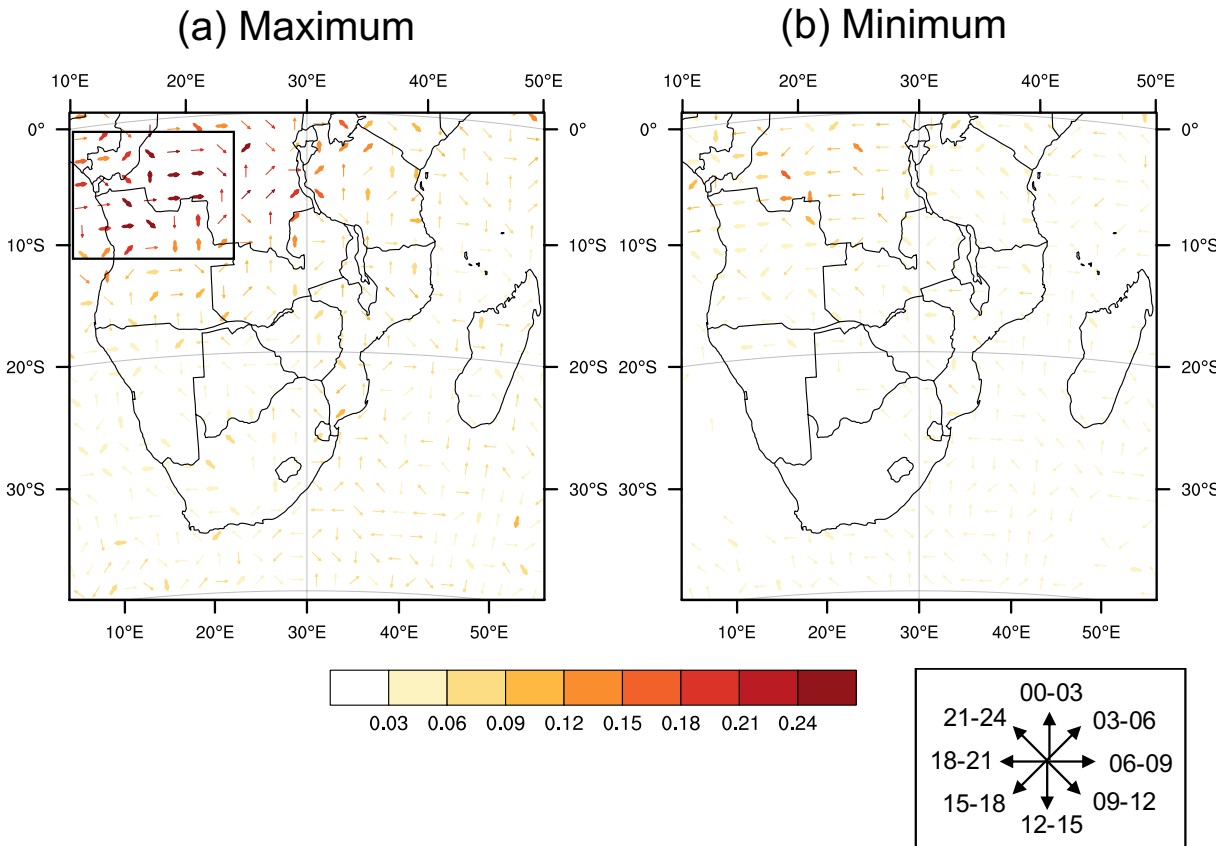

**Figure 9 Diurnally maximum and minimum DAOD across South Africa. Figure elements are the same as in Figure 5. The box indicates the tropical southern Africa.**



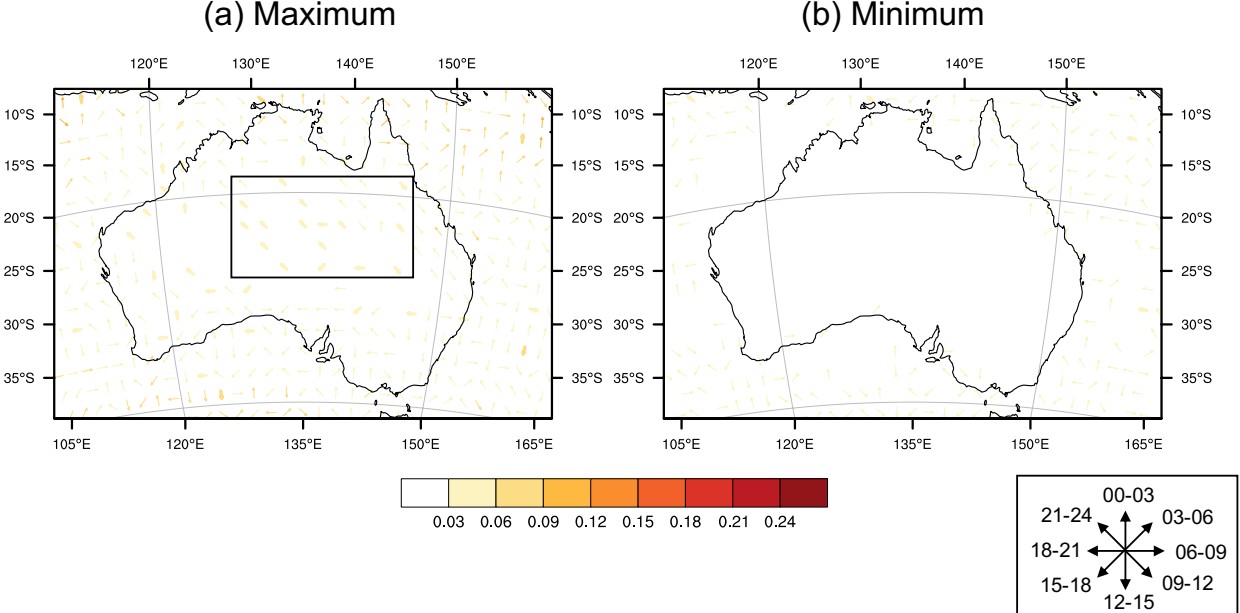

**Figure 10 Diurnally maximum and minimum DAOD across Australia. Figure elements are the same as in Figure 5. The box indicates central Australia.**