# Peer review of "A Global Analysis of Dust Diurnal Variability Using CATS Observations"

_Atmospheric Chemistry and Physics, 2019_

## Referee Comment (RC1) · Anonymous Referee #2 · 5 Dec 2019

General comments:

Characterizing any possible diurnal variation of different aerosol species in the atmosphere on a global basis is clearly an important project and can only be achieved by satellite measurements. However because of orbital restrictions and coverage issues most current satellite instruments lack this capability. CATS lidar that flew on the International Space Station between February 2015 and October 2017 was the only one which could provide diurnal information after suitable aggregation of the data and this has been one of the main planks of the CATS mission. This paper attempts to characterize the diurnal variability of dust globally using CATS data. The introduction is well written and lays out the scope and objectives of the paper in a structured way. However the analysis that follows does not provide evidence of diurnal variability of

dust from these measurements in a clear manner. In fact some of the aerosol diurnal variability using CATS data were already presented in Lee et al. (2019) which included North Africa and Middle East with dust being the dominant species. However, the very recent paper by Pauly et al. (2019) point out the high daytime lidar calibration uncertainty at 1064 nm (16-18%) with a corresponding uncertainty of ∼21% in daytime total attenuated backscatter, which is significantly larger than the uncertainty in the nighttime total attenuated backscatter at the same wavelength (∼7%). The day and night extinction profiles and optical depths will be impacted by these differences. Any meaningful discussion of diurnal variation in CATS aerosol data should clearly address this issue. The authors discussed the day/night data issue by comparing with AERONET data, which is not convincing (see below) and do not even mention the daytime calibration issues as discussed in Pauly et al. (2019). The presentation of the diurnal variability using time vectors is interesting but does not clearly establish the full diurnal variability. The vertical information content from the lidar measurements in characterizing the diurnal variability has also been ignored. I was also frustrated by the frequent references to "hypothetical" meteorological drivers despite the claim of "underlying meteorological processes are discussed in detail". Overall, I am not quite convinced by the analysis and regret that I am unable to recommend publication of this manuscript in its present form in ACP.

Specific comments:

1. More details should be given about the CATS data for the sake of completeness and for the reader who may not be initiated into the lidar terminology. For instance define the depolarization ratio and meaning of the various QA terms. How is "dust" classified and what does "dust mixture" refer to—is the latter like the "polluted dust" in CALIPSO terminology? It might even be nice to present a browse image of a dust plume as captured by CATS and/or an extinction profile to set the context.

2. I am a bit surprised that the authors do not attempt to use the CALIPSO lidar data in their analysis. Dust retrieval is probably one of the best products from CALIPSO

measurements. Even if CALIPSO reports only at 1:30 and 13:30 hrs local time, it should be useful to compare the CATS dust profiles at those local times. This was done, for instance, by Noel et al. (2019) in their study of cloud diurnal variation using CATS data.

3. More details of the figures should be given in the text. For instance, it is not clear if the supplemental Figure 1 (also Figure 4) is for a specific year or climatology. Similarly the presentation of maximum/minimum DAOD in Figures 5-10 could be clarified in the text and not just in the caption to the figures.

4. The comparison between CATS and AERONET data in Figure 1 is intriguing. As the authors themselves point out, the nighttime AERONET data are not quality assured. The sample size in the nighttime is quite low compared to the daytime. In particular, the nighttime sampling is very sparse over the dust belt and this paper is eventually concerned with the dust diurnal variability. As mentioned above, the significantly lower SNR in the daytime data and the high uncertainty in the daytime calibration (Pauly et al., 2019) are issues that should be clearly addressed in the context of day/night data quality and how they impact the diurnal variability of DAOD.

5. I do not understand the point of presenting Figure 3. The authors simply show 550 nm DAOD from MISR and 1064 nm DAOD from CATS without any attempt to convert the two datasets to the same wavelength, even if the purpose is to compare only the general spatial pattern. In a similar study using CATS aerosol data, Lee et al. (2019) had converted the MODIS data to 1064 nm using an Angstrom exponent. Once again this analysis could be done using the CALIPSO data as well. Besides, the correlation coefficients (less than 0.4 in all seasons) hardly bolster the authors' argument. This figure is essentially a comparison of daytime data and once again the data quality issues come to mind. I am also curious as to why CATS and MISR both show significant dust plumes over the South Atlantic region in the biomass burning season in southern Africa (JJA/SON)? If this dust was generated over the land, I would have expected to see dust also over the source regions in the land and a corresponding gradient from

land to ocean. Is there a scope for misclassification of smoke by any chance? Why do we see so much dust at the highest southern latitudes in the MISR data in SON (less so in the CATS data)?

6. The color bar used for Figure 4 does not show any dust loading over the north western US during the majority of the time windows, presumably because DAOD is less than 0.1 – this can confuse things as the authors later discuss diurnal variability over this region. Similarly I do not see much dust loading over Australia in this Figure. A different color bar should be used.

7. Figures 5-10 are presented as evidence of the diurnal variability in various regions of the globe with the vectors giving the times of maximum and minimum dust loading. This seems like an interesting way to present this, but not particularly convincing—often they show multiple times of maximum within the same box and do not provide a sense of the full diurnal variability in a quantitative way. The boxes should be labelled within the plots for easy readability and given for both maximum and minimum plots. In fact a regular plot showing the DAOD as a function of the local time and a discussion vis-a-vis the day/night difference in data quality would be more convincing. In all cases, some supporting evidence from primary meteorological drivers should be presented in a quantitative way, rather than simply hypothesizing. In Figure 6 the box over Thar desert seems to cover much of central India including parts of the Indo Gangetic basin, which is misleading. What about the Gobi desert?

8. I believe this paper can be improved by delineating the diurnal variation in different seasons. For instance, this might reveal any diurnal variation over the Thar desert area during the boreal summer monsoon season when local dust sources dominate rather than transported dust from West Asian sources, as stated in section 3.3.2. As well, the vertical information available from CATS lidar should be exploited to discuss the altitude of maximum diurnal variation (e.g. Lee et al., 2019).

---

## Referee Comment (RC2) · Anonymous Referee #1 · 20 Dec 2019

The paper "Global Analysis of Dust Diurnal Variability Using CATS Observations" by Yan Yu et al. investigates the diurnal cycle of dust loading across the global tropics, sub-tropics, and mid-latitudes by analyzing aerosol extinction and typing profiles observed by CATS lidar aboard the ISS. CATS was developed to address three main science objectives; with one of the goals to measure and characterize aerosols/clouds on a global scale and at various local times. The diurnal variability of aerosols consists a significant scientific question partially addressed until recently mainly based on sunphotometers (e.g. Smirnov et al., 2002) and ground-based lidar systems (e.g. EARLINET; Pappalardo et al., 2014), important for a large number of applications / impacts (radiative forcing, aerosol-cloud interaction, public health). Until CATS, the high importance of dust (Kok et al., 2012) was studied on a large scale over the dust sources

based mainly on geostationary satellites (e.g. MSG-SEVIRI; Schepanski et al., 2007). The present study attempts to build on the aerosol diurnal study performed by Lee et al., 2018, focusing on dust aerosols. The idea of the study is of high scientific interest, falls within the scope of ACP, the manuscript is well-written / structured, the presentation clear, the language fluent. However, despite the significance of the scientific idea, the performed approach and methodology are subject to major deficiencies and the results are rather questionable.

Here are some of my main comments which I think will help the authors to improve their manuscript. 1) The authors have established the diurnal variability of dust over main dust source regions based on the concept that there is insignificant difference between CATS daytime and nighttime observations. Some indicative examples are: "... there is no significant difference between daytime and nighttime CATS AOD quality" – line 176. "The currently identified insignificant difference between daytime and nighttime CATS data quality is hypothesized to be partly attributed to ..." – line 181. "Although there is no significant difference between the daytime and nighttime CATS data quality ..." – line 188. "... According to the comparison with ground-based and other satellite observations, CATS aerosol and dust loading observations exhibits reasonable quality and insignificant day-night inconsistency" – Abstract. However this assumptions/hypothesis is not valid. Pauly et al. (2019) extensively addressed the calibration and performance of CATS L1B - ATB based on comparisons with CPL, CALIOP (CCAVE) and PollyNET observations and reported on the significant nighttime and daytime differences. Similarly, Proestakis et al. (2019), implemented a large number of EARLINET stations and collocated ISS-CATS observations and reported on the performance of CATS backscatter coefficient, including on the significant underestimation in CATS daytime observations. The aforementioned studies build on the already reported by Yorks et al., (2016) minimum detection thresholds, with CATS in the case of nighttime to be approximately two orders of magnitude more sensitive than during daytime. Similarly, other studies have reported on the performance of CATS (e.g. Lee et al., 2019; Rajapakshe et al., 2017, Noel et al., 2018). The daytime underestimations compared

to nighttime observations is an issue which is not considered properly, it is ignored as not significant, and not addressed, discussed properly and eventually resulting in questionable conclusions. 2) The authors study the diurnal variability of dust based on CATS aerosol subtype classification. To be more specific, the authors state that: "... DAOD is defined here as the vertical integral of aerosol extinction coefficient over "dust" (Aerosol Type = 3) or "dust mixture" (Aerosol Type = 4) ...". The subtype classification algorithm depends on inputs, one of them the depolarization. 1064nm dust linear depolarization ratio vary strongly between 0.22 and 0.28 to 0.4 (e.g. Freudenthaler et al., 2009; Burton et al., 2015; Haring et al., 2017). On the other hand this has an effect to the non-dust component in "dust" and "polluted-dust" aerosol types considered in the study, always present when the depolarization is lower than 0.22. Although there are methodologies developed to address the decoupling of dust and non-dust components (e.g. Tesche et al., 2009; Mamouri and Ansmann 2014; Amiridis et al., 2013), the authors have performed a more bulk approach, reporting the diurnal variability not only of the dust component but the non-dust component as well, contaminating the results. 3) Although the ISS inclination is confined between (approximately) 51oS and 51oN, there is a clear significantly larger number of ISS overpasses/CATS observations over the Saharan Desert during nighttime that during daytime. The authors should clearly present the available sample of observations per 3-hour over each selected region, and discuss the sample effect. 4) No vertical mean extinction coefficient profiles (including statistical indicators such as SD) have been included, a significant advantage of lidar system compared to passive sensors. In addition, although CALIOP does not provide observations at various local times, since CALIPSO is the longest existing lidar system is space, observations at least during the overpass times should be included.

Considering the above comments, I suggest to ACP journal to reject the paper. The authors should go through the entire manuscript more carefully before resubmitting it.
* * *

---

## Referee Comment (RC3) · Anonymous Referee #3 · 20 Dec 2019

The investigation proposed by Yu et al. is very promising: the diurnal variability of the dust around the globe is an essential information for many scientific fields and for the variety of dust impacts and potential applications. Nevertheless, I would suggest a major and careful revision of the paper which should take into account the comments of the other 2 reviewers. Additionally, from my side I want to underline the following as points to which authors should pay attention:

- CATS daytime measurements are demonstrated to have some issues and are less accurate than the nighttime ones. Using daytime and nighttime info in the analysis should take into account this.

- It is not clearly reported why the authors used CATS data and not CALIPSO ones. Is there any advantage in CATS data for doing so? I think yes, but the authors should

stress it more.

- As the other reviewers pointed out, the vertical information is completely missing here: lidar big advantage is the profiling capability and authors simply disregarded it. This is really frustrating

- But more frustrating is expecting to see information about the variability and not getting it at all from the paper: the results of the paper are about where the variability is observed and significant (the exact meaning of this term is not fully clear here). How much DOAD change in 3 hours? Not a number about this for the interested regions. Please reconstruct the results and discussion part reporting really the variability information.

Finally, I found really strange the extremely low values observe in Australia (DAOD from Ridley et al., ACP 2016 is around 0.005-0.01!). Is there any problems for not tropics regions in CATS data/algorithms?

—————————————————